# Physical Exercise and the Hallmarks of Breast Cancer: A Narrative Review

**DOI:** 10.3390/cancers15010324

**Published:** 2023-01-03

**Authors:** Celia García-Chico, Susana López-Ortiz, Saúl Peñín-Grandes, José Pinto-Fraga, Pedro L. Valenzuela, Enzo Emanuele, Claudia Ceci, Grazia Graziani, Carmen Fiuza-Luces, Simone Lista, Alejandro Lucia, Alejandro Santos-Lozano

**Affiliations:** 1i+HeALTH, Miguel de Cervantes European University, 27038 Valladolid, Spain; 2Research Institute of the Hospital 12 de Octubre (‘Imas12’ [PaHerg Group]), 28041 Madrid, Spain; 3Department of Systems Biology, University of Alcalá, 28871 Madrid, Spain; 42E Science, Robbio, 27038 Pavia, Italy; 5Departmental Faculty of Medicine, Saint Camillus International University of Health and Medical Sciences, 00133 Rome, Italy; 6Department of Systems Medicine, University of Rome Tor Vergata, 00133 Rome, Italy; 7Faculty of Sport Sciences, Universidad Europea de Madrid, 28670 Madrid, Spain

**Keywords:** breast cancer, mammary cancer, physical activity, training, biomarkers

## Abstract

**Simple Summary:**

The growing prevalence of breast cancer, together with the progress in updating tumor hallmarks, increases the need to develop and investigate the molecular pathways that influence the progression of the disease. It is known that lifestyle greatly influences the disease onset and prognosis, but no research has yet been carried out that synthesizes this relationship in depth. The present narrative review aims to describe the effects of physical exercise on breast cancer hallmarks.

**Abstract:**

Growing evidence suggests that, among the different molecular/cellular pathophysiological mechanisms associated with cancer, there are 14 hallmarks that play a major role, including: (i) sustaining proliferative signaling, (ii) evading growth suppressors, (iii) activating invasion and metastasis, (iv) enabling replicative immortality, (v) inducing angiogenesis, (vi) resisting cell death, (vii) reprogramming energy metabolism, (viii) evading immune destruction, (ix) genome instability and mutations, (x) tumor-promoting inflammation, (xi) unlocking phenotypic plasticity, (xii) nonmutational epigenetic reprogramming, (xiii) polymorphic microbiomes, and (xiv) senescent cells. These hallmarks are also associated with the development of breast cancer, which represents the most prevalent tumor type in the world. The present narrative review aims to describe, for the first time, the effects of physical activity/exercise on these hallmarks. In summary, an active lifestyle, and particularly regular physical exercise, provides beneficial effects on all major hallmarks associated with breast cancer, and might therefore help to counteract the progression of the disease or its associated burden.

## 1. Introduction

The World Health Organization defines cancer as a large group of diseases arising in almost any organ or tissue of the body wherein abnormal cells are subject to uncontrolled growth, go beyond their usual boundaries to invade adjacent parts of the body, and/or spread to other organs [1]. The factors promoting tumor proliferation, designated as “cancer hallmarks”, have been previously described in detail [2,3,4] and include: (i) sustaining proliferative signaling, (ii) evading growth suppressors, (iii) activating invasion and metastasis, (iv) enabling replicative immortality, (v) inducing angiogenesis, (vi) resisting cell death, (vii) reprogramming energy metabolism, (viii) evading immune destruction, (ix) genome instability and mutation, (x) tumor-promoting inflammation, (xi) unlocking phenotypic plasticity, (xii) nonmutational epigenetic reprogramming, (xiii) polymorphic microbiomes, and (xiv) senescent cells [2,3,4].

Breast cancer is recognized as the most prevalent type of cancer worldwide [1]. Specifically, it refers to a type of cancer that originates in breast tissue, usually in the mammary lobules and milk ducts [5,6,7] and involves pathological conditions showing a high degree of heterogeneity at both the molecular and cellular levels. This pathology is associated with a longer survival rate than other types of cancers [5]. However, frequent recurrences, disease progression to metastasis, and surgery-associated disorders [5] compromise both the physical and mental integrity of patients. 

Although the underlying causes of breast cancer remain unknown [6], there is growing evidence highlighting the potential impact of certain risk factors, both extrinsic and intrinsic, on its pathogenesis. For instance, following a healthy lifestyle, particularly engaging in regular physical activity or physical exercise, might reduce the risk of developing the disease [5,8,9,10,11,12,13,14] as well as improve its prognosis in already affected patients [9,15,16]. A basic distinction is required in this context where physical activity is considered as any bodily movement involving skeletal muscles through energy consumption, while physical exercise refers to a subcategory of planned, structured, and repetitive physical activity, the aim of which is to improve or maintain one or more features of physical fitness [17].

Physical activity is related to reduced breast cancer mortality and recurrence in breast cancer patients [18,19], as well as fewer/less severe adverse effects following its treatment [20]. Unfortunately, women with a diagnosis of breast cancer tend to reduce their physical activity levels by 11%, and an even greater decrease has been observed in patients who are treated with chemotherapy (50%) and radiotherapy (24%), compared to untreated patients [21]. Since physical exercise is of vital importance in improving physiological processes, like cardiorespiratory fitness, muscular strength, and psychological wellbeing, and considering that a discernible reduction in physical performance, a negative change in body composition (e.g., increase in body mass), and an increased tendency toward depression or anxiety are common side effects of cancer treatment [22], the role of exercise after breast cancer therapy has become an important area of research. About one third of breast cancer patients treated with adjuvant chemotherapy suffer an impairment of their cardiorespiratory fitness [23], and therefore the role of exercise training to mitigate the cardiotoxic effects of chemotherapy deserves further investigation. Relatively few data are available, from studies aimed at identifying the most effective aerobic training protocol, in terms of exercise modality, timing, and duration, to increase the cardiorespiratory fitness and to lower the risk of cancer-related mortality [24,25]. For instance, a recent study reported that aerobic exercise exerts a beneficial effect on the cardiotoxicity induced by anthracyclines and by the anti-HER2 monoclonal antibody trastuzumab in women with breast cancer, improving diastolic function and cardiorespiratory fitness [26].

Our research group has previously summarized the potential beneficial effects of exercise training on cancer hallmarks [27]. In addition, a systematic review summarized the effects of physical exercise on certain breast cancer biomarkers [24]. However, to the best of our knowledge, no review has previously explored how exercise specifically affects breast cancer hallmarks. Therefore, the primary aim of the present narrative review is to summarize the effects of physical exercise on cancer hallmarks, after their specific contextualization in the field of breast cancer according to both preclinical and clinical studies. 

## 2. The Effects of Physical Exercise on the Hallmarks Associated with Breast Cancer

### 2.1. Sustaining Proliferative Signaling 

As their primary incontrovertible feature, cancer cells are able to modify the signals that regulate cell growth and division [3]. Some hormones, such as estrogen or progesterone, and growth factors, such as human epidermal growth factor, by acting on human epidermal growth factor receptor 2 (HER2), participate in cell proliferation signaling [28,29,30]. Alterations in these pathways induce molecular and biological changes, facilitating the establishment of the disease [31].

Although a large amount of evidence comes from animal models [32,33,34,35,36,37], data are also available from clinical evaluation of the effects of physical exercise and physical activity on circulating serum markers related to cancer cell proliferation [38]. 

In detail, preclinical data from male mice show that anaerobic exercise training for eight weeks (four sessions per week) reduces both tumor weight and cell proliferation ex vivo when compared to a sedentary group [37]. Likewise, aerobic exercise seems to affect certain variables associated with cell proliferation in a transgenic mouse model of breast cancer and N-methyl-N-nitrosourea (MNU)-induced mammary carcinogenesis in female rats, including the reduction in tumor size [32], the number of developed tumors [33], or the expression of proliferation markers, such as the Ki-67 nuclear protein [34]. However, there are discrepancies regarding the effect of exercise in the expression of Ki-67, since in some cases its expression is not significantly different between exercised and not exercised animals or is not paralleled by changes in training intensity [32,33]. Aerobic exercise also reduces the expression of both estrogen and progesterone receptors in animal models, thus further acting as a protective factor against tumor proliferation and development [36]. 

Another mechanism promoting cell growth and proliferation lies in the binding of insulin-like growth factor-1 (IGF-1) to the insulin-like growth factor receptor type 1 (IGF-1R). Indeed, high levels of IGF-1R are associated with a worse disease prognosis in breast cancer patients, as assessed through a computational analysis of cancer genomics information available in public databases [39]. Various pathways activate the aforementioned process, thus stimulating tumor survival [39]. In the only available clinical studies evaluating tumor proliferative signaling markers, combined training, based on resistance plus aerobic exercise, reduces serum IGF-1, estradiol, and insulin levels [40] in breast cancer survivors who are overweight or obese [38]. Additionally, moderate-intensity aerobic exercise decreases both insulin and IGF-1 levels [41]. 

Another cell-cycle specific antigen and proliferation marker in breast cancer is proliferating cell nuclear antigen (PCNA), having an important role in DNA replication [42]. Exposure of human breast cancer cell lines to an exercise-conditioned medium obtained from contracted myotubules, with or without doxorubicin, reveals a reduction in PCNA expression and a synergistic relationship with the anticancer drug [43] (Figure 1).

### 2.2. Evading Growth Suppressors 

Besides preserving cell proliferation, tumor cells are able to evade the mechanisms that, under physiological conditions, regulate cell growth and proliferation [3].

Tumor suppressor genes and oncogenes are the primary genes related to cancer development and evolution [44]. Tumor suppressor genes, such as tumor protein p53 (*TP53)*, retinoblastoma (*Rb*), phosphatase and tensin homolog (*PTEN*) [45], and breast cancer type 1 and type 2 (*BRCA1/2*) genes [46] are inactivated in breast cancer. On the other hand, some oncogenes, including *IGF-1R* gene, epidermal growth factor receptor (*EGFR*) gene, and oncogenic microRNAs (miRNAs) (namely, oncomiR, e.g., oncomiR-21, oncomiR-1207-5p, oncomiR-492, and oncomiR-135b) [47,48] are overexpressed. Moreover, the “Hippo” tumor suppressor signaling pathway, which regulates cell growth, may be dysregulated in breast cancer patients [49].

Excessive p53 protein activation increases insulin-like growth factor binding protein-3 (IGFBP-3) [50] and PTEN protein expression [51], which may lead to a downregulation of the IGF-1 pathway, thus promoting an antiproliferative and antimetastatic effect. Conversely, the loss of tumor suppressor genes, such as *TP53*, *PTEN*, and *BRCA1*, contributes to an increase in IGF-1R protein expression in tumors [52].

IGF-1R protein, overexpressed and hyperphosphorylated in many breast cancer subtypes [53], contributes to invasion and metastasis via an increase in the proliferation rate of tumor cells and a decrease in the rate of their destruction [54]. In detail, IGF-1R activates the phosphoinositide-3 kinase/protein kinase-B/mammalian target of rapamycin (PI3K/Akt/mTOR) intracellular signaling pathway, which promotes cell proliferation and inhibits programmed cell death [53]. This signaling pathway is also related to the repression of *PTEN* and *BRCA1* [49]. Furthermore, the other oncogene, *EGFR*, interacts with the same PI3K/Akt signaling pathway [47]. 

Aerobic exercise increases the *TP53* gene levels in female mice with triple-negative breast cancer (TNBC), despite no relative change in PTEN expression [55], and reduces the expression of hyperphosphorylated Rb protein (inactive form), in rats with mammary adenocarcinoma [56]. A combination of aerobic and resistance exercise increases the serum levels of IGBP-3 in overweight or obese breast cancer survivors [38]. Moreover, exercise influences the expression of some miRNAs that act as tumor suppressors. Preclinical and clinical evidence shows that acute and chronic exercise interventions raise the expression of miR-133, which alters cancer progression by acting on oncogenes *IGF-1R* and *EGFR* [47]. Additionally, high-intensity interval training (HIIT) upregulates the expression of miR-206 and miR-143 in women with breast cancer [57]. Evidence on the role of myokines, i.e., cytokines, peptides, or growth factors produced and released by skeletal muscle cells under contraction, is still limited in breast cancer. Available research shows that among myokines, aerobic exercise releases oncostatin M (OSM), thus resulting in lower tumor volume in breast cancer mice [58]. Breast cancer patients show significantly decreased serum irisin concentrations [59]; hence, it is reasonable to assume that physical exercise, by stimulating the production of irisin, activates the suppression of cell growth, as in other types of cancer (i.e., prostate cancer and glioblastoma) [47]. Moreover, irisin significantly decreases cell number, migration, and viability of malignant MDA-MB-231 breast epithelial cells in vitro, without affecting nonmalignant MCF-10a cells, and enhances the cytotoxic effects of doxorubicin [60].

Finally, exercise produces epinephrine and norepinephrine that activate the Hippo pathway, thus inhibiting breast cancer cell growth and the mTOR pathway [61]. 

In general, existing studies largely support the claim that physical exercise activates tumor suppressors and inactivates tumor promoters by hampering the establishment of a favorable microenvironment for cancer progression (Figure 1). 

### 2.3. Activating Invasion and Metastasis

Tumor complexity is related to its microenvironment (tumor microenvironment, TME), the composition of which differs among tumor types and breast cancer subtypes [62,63]. TME comprises the proliferating tumor cells along with a range of noncancerous cells (i.e., stromal cells) incorporated in the tumor mass [62]. The noncancerous component of the TME includes immune cells, blood vessels, the extracellular matrix, and stromal cells [63]. Macrophages are stromal cells in TME, wherein they differentiate to become tumor-associated macrophages (TAMs). TAMs can polarize into two phenotypes, namely M1 (showing antitumor effects) and M2 (supporting tumor growth, local invasion, metastasis, and exerting an immune-suppressive function) [64]. In this context, TME affects angiogenesis, proliferation, invasion, and metastasis through the release of growth factors and cytokines [63]. 

The breast adipose tissue, mainly consisting of adipocytes, constitutes the breast stroma, which generates high concentrations of cytokines and adipokines that participate in the paracrine crosstalk with cancerous cells and contribute to increased cell proliferation and invasion [65,66]. 

Some cytokines and adipokines, such as leptin, the previously mentioned irisin (also defined as an adipo-myokine), and OSM, as well as resistin, are involved in metastasis and invasion in breast cancer [67]. The first process implicated in metastasis is the epithelial-to-mesenchymal transition (EMT) [68], characterized by the loss of epithelial markers (e.g., E-cadherin) and the appearance of mesenchymal markers [69]. In this context, resistin [70] and EGF [69] enhance the metastatic potential of cancer cells by promoting the EMT. The activation of the TGF-β [71] and hypoxia-inducible factor 1 (HIF-1) signaling pathways can also promote the EMT through several mechanisms [68] and upregulate the activity of matrix metalloproteinases (MMPs, particularly MMP-2 and MMP-9), which degrade components of the extracellular matrix, promoting the invasion and intravasation of cancer cells in the blood vessels and their traveling through the systemic circulation toward distant sites [68]. 

Chronic adaptations to moderate-intensity aerobic exercise seem to prevent the activation of a metastatic cascade and the resulting formation of cancer metastasis [72] by controlling angiogenesis, eliminating circulating tumor cells, and reducing endothelial cells’ permeability, while high-intensity physical exercise seems to favor cancer spreading, likely due to disproportionate stress. 

Studies on animal models do not report significant results: in a rat model of breast cancer, moderate-intensity exercise training tends to protect from pulmonary metastases, but the small sample evaluated does not allow to reach statistical significance [73]. Conversely, a preclinical study conducted in mice show that four weeks of wheel running after breast cancer cell injection is associated with a higher number of pulmonary metastases compared to sedentary controls, as well as with a lower nitric oxide production [74]. By contrast, a pilot clinical trial in breast cancer patients shows that aerobic exercise training for 12 weeks combined with neoadjuvant chemotherapy promotes higher nitric oxide production and lower tumor invasiveness [75]. 

In any case, physical exercise helps TAMs polarize to an M1 phenotype, enhancing their antitumor effects. Additionally, moderate-to-high intensity exercise decreases the macrophages’ recruitment to the TME of breast cancer [64]. In contrast, sedentary behaviors result in M2 phenotype polarization, thus promoting tumor growth, invasion, and metastasis [64]. Physical exercise induces the release of myokines related to tumor invasion and metastasis, such as irisin and OSM. As mentioned above, irisin levels are decreased in breast cancer patients [59,76], but physical exercise can improve irisin serum levels to prevent spinal metastasis [77]. In this view, physical exercise induces the release of antimetastatic adipokines (e.g., adiponectin) [78] and, consequently, the downregulation of prometastatic adipokines, such as leptin [79] and tumor necrosis factor-α (TNF-α) [80]. Exercise also upregulates myomiR-133a, i.e., a member of a subset of miRNAs specific to striated muscle and expressed at higher levels in skeletal muscle [47], which influences myoblast differentiation and contributes to the suppression of several tumors, including breast cancer, by controlling tumor invasiveness and metastasis [47,48].

An acute bout of aerobic exercise stabilizes hypoxia-inducible factor 1-α (HIF-1α) after a period of regular endurance training through the expression of negative regulators, as a mechanism of adaptation of skeletal muscle; this long-term endurance exercise downregulation of HIF-1α response [81] could possibly decrease EMT promoted by this protein. 

Chemokine (C-X-C motif) receptor type 4 (CXCR4) and its ligand, chemokine (C-X-C motif) ligand type 12 (CXCL12), are relatively highly expressed in invasive breast cancer cells, homing them to distant lymph nodes, the lungs, and the liver [82]. In contrast to the previously discussed results, *CXCR4* gene expression is increased in a transgenic mouse model of breast cancer with access to a wheel, compared with nonrunners, although pulmonary metastatic foci and the proliferative index, as indicated by morphometric analyses and measurement of Ki-67 levels, respectively, show no differences between runners and nonrunners, and no difference has been observed in the expression of *CXCL12* gene [32] (Figure 2).

### 2.4. Enabling Replicative Immortality 

Cancer cells acquire endless replicative immortality to develop macroscopic tumors [3] via the activation of telomerase or a DNA homologous recombination-based mechanism (alternative lengthening of telomeres, ALT) [83]. Telomere shortening following each cell division is related to genomic instability, oxidative stress, and inflammation [84]. Telomerase activation restores genomic stability and enhances tumor progression by preserving the telomeric length despite cell division [27]. 

Physical exercise affects cancer development and its dependence on genetic alterations of chromosomes, DNA methylation, miRNA expression, and changes in chromatin structure [85,86]. Furthermore, exercise attenuates telomere attrition and helps maintain a balance with oxidative stress and inflammatory status [84]—factors inversely correlated with telomere length.

Related to the role of miRNAs, both acute and chronic exercise interventions increase the release of the senescence-associated myomiR-133 into circulation [47], thus supporting the inhibition of telomerase expression [87,88].

The evidence that telomere lengthening decreases mortality in women with breast cancer [89] further supports the hypothesis that regular physical exercise represents a potential protective factor against this tumor hallmark (Figure 3).

### 2.5. Inducing Angiogenesis 

The formation of a hypoxic microenvironment is a well-known event in the development of mutagenesis and cancer [90]. Hypoxia maintains stem cells in an undifferentiated state, enabling only cancer cells to differentiate and accumulate genetic and epigenetic alterations uninterrupted [90]. As the tumor mass increases in size, the oxygen availability decreases. In response to hypoxia, the development of new blood vessels from the pre-existing vasculature, i.e., angiogenesis, occurs to obtain the nutrients and oxygen required for massive cell growth, as well as to promote malignant cells entering the systemic circulation, thus resulting in metastasis [91]. In this context, cancer’s capacity to progress is endless, due to its ability to sustain blood supply [92]. Both the tumor itself and the stromal cells of the hypoxic TME produce and secrete angiogenic growth factors and upregulate their receptors. The activation of pro-angiogenic pathways causes endothelial cells’ proliferation and migration and leads to an increased vascular permeability [90], thus making angiogenesis a key process for cancer cells’ intravasation. The main hypoxia-inducible angiogenic stimulator, not surprisingly upregulated by HIF-1 at the transcriptional level, is the vascular endothelial growth factor-A (VEGF-A) [93]. In turn, the most consistently induced miRNA during hypoxia, i.e., miR-210, promotes angiogenesis by enhancing VEGF-A expression. Another miRNA related to angiogenesis is miR-21, which targets *PTEN* to activate the Akt/ERK signaling pathway, leading to a high expression of both HIF-1α and VEGF-A [94]. However, the excessive production of VEGF-A in the hypoxic TME causes the new vasculature to be structurally and functionally abnormal [95] with chaotically organized tumor vessels not resembling the hierarchical structural organization of noncancerous vascular networks.

Preclinical research in an orthotopic murine model of breast cancer shows that exercise promotes tumor vessel maturity and microvessel density [96], supported by reduction in intratumoral hypoxia and improvement in tissue perfusion, oxygen supply [97], and nitric oxide bioavailability [98], thus leading to an unfavorable TME for cancer progression and to a better drug delivery [99]. Indeed, since neovessels constitute the major route for cancer cells’ proliferation and dissemination, a process of vasculature “normalization”, through which abnormal vessels are remodeled, can be beneficial in terms of a better response to anticancer treatment [99].

Another preclinical study conducted in mice suggests that performing treadmill endurance exercise decreases VEGF-A expression, as assessed through Enzyme-linked immunosorbent assay (ELISA) from tissue samples [100], while a rat model of a mammary tumor confirms an increased VEGF-A expression, detected immunohistochemically, and tumor vascularization after long-term training [101]. In this context, a clinical study reports that an increase in the total physical exercise time and weight loss leads to a decreased circulating VEGF-A expression in overweight women with breast cancer [102] (Figure 2). However, a combined training (i.e., aerobic and resistance training) intervention, following surgery, radiotherapy, or chemotherapy does not substantiate these results [103], since serum mice levels of VEGF-A are not significantly different between exercised and control groups.

### 2.6. Resisting Cell Death 

Cancer cells acquire a characteristic resistance to apoptosis, which prevents them from dying and sustains uncontrolled proliferation [104]. The most relevant programmed cell death pathways described so far are: (i) the extrinsic (i.e., death receptor based), which receives and processes extracellular pro-death signals, and (ii) the intrinsic (i.e., mitochondrial), which senses and integrates intracellularly originated signals [3]. 

To inhibit or avoid apoptosis, tumor cells utilize a number of molecular strategies. The loss of TP53 tumor suppressor activity is the most common, as it eliminates this important DNA damage sensor from the apoptosis-inducing circuit. Tumors can also accomplish comparable outcomes by upregulating anti-apoptotic regulators (Bcl-2, Bcl-Xl) or survival signals (IGF-1/2), downregulating pro-apoptotic factors (Bax, Bim, Puma), short-circuiting the extrinsic ligand-induced death pathway, or underexpressing caspases (caspases-3, -8, and -9) [3,105]. As expected, both overexpression of anti-apoptotic factors and inactivation of pro-apoptotic factors are often correlated with poor prognosis, recurrence, and resistance to treatments [106].

Aerobic physical exercise induces apoptosis, as demonstrated in breast cancer murine models [96,107], in terms of increased expression of Bax and caspase-3 [107]; anaerobic training increases the rate of tumor cells’ apoptosis in male rats (two folds higher compared to sedentary rats, paralleled by an increase of Bax and reduction in Bcl-2 expression) [37]. High-intensity aerobic exercise is also associated with a higher expression of Terminal deoxynucleotidyl transferase dUTP nick end labeling (TUNEL)-positive cancer cells, a biomarker of apoptotic signaling, in a chemically induced rat model of breast cancer [33]. However, decreased levels of anti-apoptotic regulators (Bcl-2 and Bcl-Xl), increased expression of the pro-apoptotic factor Bax, and caspase-3 activation are preclinically evident even when low-intensity aerobic exercise is performed [35]. Thus, even though only preclinical investigations are available, exercise seems to induce manifest improvements in apoptosis and reduces breast cancer resistance to programmed cell death, without any correlation to difficulty and energy expenditure (Figure 1). 

### 2.7. Reprogramming Energy Metabolism 

Cancer cells are able to modify their metabolism by using glycolysis even under aerobic conditions (i.e., in the presence of oxygen and fully functioning mitochondria), a phenomenon known as the “Warburg effect” [108]. This process allows cancer cells to survive in situations of fluctuating oxygen supply and results in high lactate generation [109]. Moreover, the altered glucose metabolism, in terms of increased fermentation to lactate, is responsible for the typical acidic pH of the TME [108]. 

Lipid metabolism, in combination with the Warburg effect and with an enhanced glutaminolysis, is another emerging hallmark for cancer metabolic reprogramming [110]. Indeed, lipid metabolic changes have an important impact on tumor cell growth, spread, and chemotherapeutic treatment resistance [110]. Tumor cells can boost de novo lipogenesis, fatty acid absorption, and fatty acid oxidation for energy generation and lipid storage [111].

Energy consumption induced by exercise significantly affects whole body and intracellular metabolism [112], potentially contributing to alterations into glucose synthesis and utilization.

In an experimental TNBC murine model involving sedentary or trained for 12 weeks animals, physical exercise correlates with a smaller tumor mass, mitochondria with a lower respiratory rate in the state of maximum electron transport capacity, and modulated macronutrient oxidation, almost exclusively represented by carbohydrate oxidation, while the sedentary condition metabolizes both carbohydrates and lipids [55]. 

In another preclinical study, mice with breast cancer undergoing aerobic physical training show a decreased type 1 monocarboxylate transporter expression and a shift in lactate dehydrogenase (LDH) isoforms (higher LDH-B and lower LDH-A expression), compared to the control group (Figure 3). These changes are associated with a lower concentration of accumulated lactate, together with a reduced tumor mass [113]. Of note, data exist demonstrating that the hypoxic TME leads to increased LDH-A expression, the silencing of which suppresses the tumorigenicity of breast cancer [114], whereas increased lactate production correlates with LDH-B downregulation [115] (Figure 3).

Altogether, the observed metabolic changes due to the performance of physical exercise seem to maintain glucose homeostasis in face of the required energy expenditure. Therefore, exercise could help in the regulation of cellular metabolism; however, the included studies are insufficient to draw conclusions on this matter.

### 2.8. Evading Immune Destruction

Tumor cells have the ability to evade the immune system mediated destruction through several mechanisms. One possibility is the reduction of natural killer (NK) cell receptor expression: cancer cells would not be recognized by lymphocytes and NK cells. Besides, proteinase inhibitor 9 (PI-9) expression could promote mammary tumor proliferation by enhancing resistance to lymphocyte- and NK cells-mediated immune response. Tumor cells may also overexpress anti-apoptotic molecules on their surface and, thus, resist immune-mediated cytotoxicity [116]. Finally, malignant cells may actively suppress the immune response by expressing immune inhibitory ligands or receptors (“immune checkpoints”) or by inducing apoptosis of certain antitumor lymphocytes. Referring to these immune checkpoints, the binding of the programmed death-ligand 1 (PDL-1) with its programmed death-1 (PD-1) receptor provides a suppressive signal to T lymphocytes with a decrease in the immune response [117]. Moreover, tumor cells are able to induce a TME supporting their proliferation and growth, enriched in immunosuppressive cells and soluble factors. Specifically, myeloid-derived suppressor cells (MDSC) are able to disrupt the binding of major histocompatibility complex (MHC) to CD8+ T cells and induce nitration of MHC class I molecules on cancer cells, making tumors resistant to the immune system [116]. 

In light of the abovementioned issues, the benefits of physical activity in improving the efficacy of the anticancer immune response may be particularly relevant in breast cancer patients in terms of survival, recurrence, and mortality. In fact, moderate physical activity recovers the immune function by increasing the number and activity of certain cells belonging to the immune system, such as neutrophils, monocytes, eosinophils, and lymphocytes [118].

Particularly, aerobic exercise is acknowledged to significantly improve NK cell activation in treadmill-trained female rats, decrease the accumulation of myeloid-derived suppressor cells (MDSC) in the TME, as well as promote an inverse correlation between MDSC and CD8+ T cells [119]. Furthermore, aerobic exercise performed by female breast cancer survivors can enhance NK cell cytotoxic activity and thymidine uptake [120]. Concerning resistance training, it appears to help female breast cancer survivors to decrease TNF-α production by their NK and natural killer T (NKT) cells [121]. In addition, when aerobic exercise is compared to muscular resistance exercise, the latter generates greater benefits: women with primary breast cancer undergoing chemotherapy who engage in muscular resistance exercise show a lower decrease in γδ T cells and CD8+ T cells than those who engage in endurance exercise [122]. However, a recent review highlights that there is still not enough evidence to support the specific effects of exercise on NK cells [123]: in breast cancer patients undergoing surgery, chemotherapy, or radiation therapy and performing moderate aerobic performance, the cytotoxic activity of NK cells is not significantly affected [124]. 

Lastly, combined training (i.e., aerobic and resistance) shows favorable results for women with breast cancer, as it increases the percentages of CD4+ and CD69+ T lymphocytes [125] (Figure 4). Nevertheless, in another trial, the combined exercise, together with diet intervention, does not show favorable results supporting the experimental group [126]. Therefore, the outcomes related to this hallmark in breast cancer patients seem to be heterogeneous, and further research is required to elucidate whether regular exercise can improve immune function in cancer survivors.

### 2.9. Genomic Instability and Mutation

Genomic instability designates the accumulation of mutations in a genome’s stabilizing genes [127], with consequent increase in the rate at which cells acquire genomic alterations. It is considered a necessary process for carcinogenesis to begin and progress [128]. 

As previously stated, physical activity is associated with a reduction of the risk of suffering from breast cancer and, in breast cancer patients, with a reduced tendency to progress toward the more aggressive and invasive form of the disease [129]; such benefits also seem to include women with high familial and genetic risk factors [130]. Indeed, many studies are exploring the benefits of physical exercise in individuals carrying the hereditary *BRCA* gene mutations. BRCA1 and BRCA2 proteins contribute to preserving the integrity of the genome, since they participate in the cellular response to DNA damage [131]. A pilot study reports how high-intensity interval exercise, combined with muscular resistance, increases the serum concentration of BRCA1 protein following a six-week intervention period in *BRCA* gene mutation carriers, and improves the systemic antioxidative status [132] (Figure 1). Therefore, the preservation of a healthy body weight and an active lifestyle in women carrying *BRCA* gene mutations could prevent disease development; however, further studies are needed to demonstrate this correlation [131].

Despite the outcomes related to *BRCA* gene mutations, no other experimental results are currently available that relate the practice of physical exercise to genomic instability in breast cancer patients. Therefore, at present, it is problematic to draw conclusions on this hallmark; future research is needed to analyze the potential benefits of exercise in this field and to standardize the best existing treatment options.

### 2.10. Tumor-Promoting Inflammation

Inflammation is a defense mechanism of the body that, physiologically, originates in response to trauma and/or infection and subsides once the problem is resolved. However, in the case of cancer, the process becomes chronic, due in part to an excessive production of cytokines and growth factors [133]. In breast cancer evolution, low-grade chronic inflammation is crucial [134]. 

Certain studies have investigated how muscle contraction can inhibit the activity of pro-inflammatory substances in distant or proximal muscular regions, with several authors aiming to address how physical activity modifies the inflammatory environment in breast cancer patients [135]. In particular, the effects of aerobic exercise are extensively explored. 

In preclinical studies, female mice who trained on a treadmill show decreased plasma concentrations of the pro-inflammatory factors interleukin-6 (IL-6) and monocyte chemoattractant protein-1 (MCP-1) [136]. Furthermore, aerobic exercise in combination with tamoxifen and/or letrozole reduces the expression of TNF-α and increases the anti-inflammatory interleukin-10 (IL-10) serum concentrations [137]. 

Concerning aerobic exercise carried out in clinical studies, although some analyses report no significant benefits related to inflammation [120,138], other research stresses that this form of training, along with diet, helps decrease TNF-α and IL-6 serum concentrations in women with breast cancer [102]. In addition, the increase in total exercise time is associated with greater reduction in IL-6 serum levels [102]. Likewise, exercise is related to an increase in interleukin-8 (IL-8) and decrease in interleukin-2 (IL-2) and interleukin-1β (IL-1β) plasma concentrations [75]. Additionally, the implementation of HIIT seems to be effective in reducing the plasma and serum levels of various pro-inflammatory markers (TNF-α, IL-6) [139,140] and in increasing the expression of interleukin-4 (IL-4), IL-10, and adiponectin [139], all attenuating inflammatory responses. Regarding IL-6, the interpretation of clinical results is still debated: this cytokine might present an increasing tendency toward plasma accumulation [141] throughout the disease progression that, according to the results of two randomized controlled trials [142,143], would be exacerbated in the absence of exercise. At the same time, it seems that, as physical activity increases, systemic levels of IL-6 also grow. 

In the case of resistance exercise, studies do not report secondary improvements or significant benefits in breast cancer survivors [121]. Nevertheless, there is clinical evidence showing that nontraining women present higher serum concentrations of MCP-1, compared to exercising patients [103]. 

Regarding combined exercise of aerobic and muscular resistance, it has been supposed that, over time, repeated bouts of exercise result in a higher steady-state level of serum IL-6 that contributes to a reduction in chronic inflammation, thanks to the possible anti-inflammatory properties of this complex cytokine [144]. Likewise, a decrease in leptin, considered a pro-inflammatory adipokine, has been observed in women with breast cancer following a combined training [144] (Figure 2). Several studies claim that this type of training is able to affect markers such as IL-6, TNF-α, leptin, IL-8, and C-reactive protein (CRP), decreasing their plasma levels as well as increasing adiponectin [145]. However, other investigations report that this type of exercise program does not generate significant benefits [125,126] and future research should be conducted to analyze its effects on inflammatory markers in breast cancer patients. 

### 2.11. Unlocking Phenotypic Plasticity 

Cell genetic and epigenetic traits are not fixed, but dependent on the activation of microenvironmental stimuli and alterations in gene expression. Such activation allows breast cancer cells to switch between different phenotypic states, a feature referred to as cell plasticity [146]. Tumor-induced hypoxia reduces the levels of ten-eleven translocation (TET) demethylases, leading to hypermethylation [4]. Publications addressing the phenotype variability of several tumor types and the effect of exercise on these different phenotypes are presently limited. However, systemic acidosis induced by physical exercise could affect signals from the TME and prevent tumor transition from a noninvasive to an invasive phenotype [147].

An explanation for understanding the mechanism by which exercise could modulate the phenotypic variability of breast cancer and surrounding cells, such as myocytes of the pectoralis major or minor, is currently missing [148]. However, it is known that each muscle fiber contains a subset of satellite, or progenitor, cells that remain quiescent unless activated. Resistance exercise may trigger a release of inflammatory substances and growth factors, thereby stimulating satellite cells and promoting their entry into the cell cycle [148]. This stimulation, a priori beneficial in healthy individuals, requires further examination in breast cancer patients. 

Notably, an analysis in women with operable breast cancer highlights that supervised aerobic exercise can increase the number of circulating endothelial progenitor cells (CEP) [75], thus potentially enhancing angiogenesis and decreasing tumor hypoxia. Moreover, in relation to phenotypic plasticity, CEP may facilitate the acquisition of an invasive tumor phenotype and metastasis. However, further analyses examining the effects of aerobic exercise are needed to determine whether the increase in CEP significantly impacts breast cancer invasiveness.

As previously reported, physical exercise promotes the polarization of TAMs to the M1 phenotype at the expense of the M2 phenotype, the first exerting antitumor effects, the latter promoting tumor aggressiveness (Figure 2). Consistently, sedentary lifestyles result in M2 phenotype polarization, which promotes tumor growth, invasion, and metastasis [64]. 

Therefore, a close connection seems to bind two different cancer hallmarks—such as phenotypic plasticity and invasion/metastasis—also evidenced by the role played by EMT in cancer metastasis. However, at present, poor data are available and the effects of physical exercise on the plasticity of breast cancer cells and CEP are still unclear.

### 2.12. Nonmutational Epigenetic Reprogramming

Over the last 10 years, possible mutation-less cancer evolution, based on the epigenetic reprogramming of the most characteristic cancer phenotypes, has been increasingly the subject of study. Hypoxia limits blood nutrient bioavailability and enhances the malignant transformation of mammary cells [4]; accordingly, previous results demonstrate that strength exercise could increase systemic acidosis [147] and decrease tumor hypoxia [75], thus decreasing the survival conditions of TME. However, it is known that epigenetic alterations (e.g., promoter methylation and/or histone modification) could also lead to disease initiation [149] in breast cancer; data have been accumulated on the effects of exercise on this cancer hallmark.

One clinical trial has demonstrated that aerobic exercise upregulates the expression of crucial tumor suppressor genes by reducing their methylation (usually responsible for transcriptional silencing), which would correlate with increased survival in breast cancer patients [150]. In particular, lethal(3) malignant brain tumor-like 1 (*L3MBTL1*) is an example of a tumor suppressor gene whose methylation increases in the control group and decreases in the exercise group. High *L3MBTL1* expression leads to a 60% reduction of the risk of breast cancer death, compared to low expression levels. Furthermore, low-grade tumors show higher *L3MBTL1* expression than high-grade tumors. Therefore, exercise apparently induces favorable effects on DNA methylation by reducing the methylation of the tumor suppressor gene *L3MBTL1* and increasing its expression (Figure 3) [150]. 

Another clinical investigation reports the significant impact of promoter methylation of certain genes, namely cyclin D2 (*CCND2*), twist family bHLH transcription factor 1 *(TWIST1*), adenomatous polyposis coli (*APC*)*,* and high in normal-1 (*HIN1*), on survival after breast cancer diagnosis, and the inverse relationship between physical activity before diagnosis and breast cancer mortality. Indeed, mortality is lower only among physically active women with promoter methylation of *APC*, *CCND2*, *HIN1*, and *TWIST1* in tumors, but not in those with unmethylated tumor markers. At the same time, no significant association between physical activity and overall methylation has been observed [151].

Oxidative stress is another molecular mechanism assumed to be responsible for epigenetic alterations preceding cancer development, which could increase the methylation of tumor suppressor genes and, as shown in the aforementioned study [150], worsen the clinical situation of breast cancer patients. Exercise can act as an antioxidant, unless it is performed extensively in unprepared tissues, where it may increase the production of reactive oxygen species (ROS) and lead to epigenetic alterations [152]. 

Although studies are available on the effects of exercise on the methylation of certain genes, the scientific literature on this topic is limited and exercise-dependent epigenetic alterations require further investigation. The potential effects of exercise on oxidative stress should also be explored to determine whether it effectively modifies the methylation status of certain genes.

### 2.13. Polymorphic Microbiomes

Current scientific evidence is trying to describe the ability of the gut microbiome to confer susceptibility to various types of cancer and its impact on therapeutic response [153]. Specifically, for breast cancer, the presence in greater abundance of *Mobiluncus*, *Brevundimonas*, *Actinomyces* [154], and *Fusobacterium nucleatum* [155] has been reported. The latter has been correlated with an increase in tumor progression, metastasis, and the inhibition of antitumor immunity through the decrease in CD4+ and CD8+ T cells [155].

Lifestyle, including dietary patterns and physical activity, is known to substantially influence the composition of the microbiota [156]. Therefore, the effects that exercise performance provides on this hallmark of cancer are increasingly gaining importance. Indeed, physical exercise is able to alter the composition of the gut microbiota and the resulting metabolic products, by reducing inflammatory signaling pathways usually induced by being overweight or obese [153]. 

It is known that diet affects the gut microbiota, and that physical activity provides a positive effect on intestinal motility, as it reduces transient deposition time and contact time between pathogens and the gastrointestinal mucosal layer [118]. Interestingly, however, exercise can beneficially modify the gut microbiota independently from diet. Observational studies indicate that physical activity correlates with a higher ratio of Firmicutes to Bacteroidetes, in turn associated with higher oxygen consumption; women who exercise a minimum of three hours per week have a higher abundance of butyrate-producing bacteria and *Akkermansia muciniphilia* (AM), a condition that correlates with a lean body mass index [157,158]. Consistently, among women with early-stage breast cancer, body composition is associated with AM growth and microbiota diversity: women exhibiting a higher amount of body fat present a lower enrichment of AM [159], while higher levels of AM may be associated with a decrease in body fat, which could translate into an improvement in the prognosis and progression of the disease.

Another hypothesis relating the carcinogenesis of breast cancer to the intestinal microbiota is microbiota role as a principal regulator of estrogen metabolism. [160] Evidence exists of a correlation between high levels of physical activity and a decrease in the concentration of the sex hormones that play a major role in promoting the proliferation of neoplastic breast epithelium [153].

Likewise, a pilot study exploring the correlation between microbiota composition and alterations in cardiorespiratory fitness and psychosocial outcomes in breast cancer survivors, reports an association between fatigue interference and microbial diversity [161]. Moreover, an association has been observed between gut microbial composition and cardiorespiratory fitness. Therefore, exercise could affect the microbiota-gut-brain axis, promoting favorable changes in certain measures, such as fatigue and cardiorespiratory fitness [161] (Figure 4).

Apart from these analyses, currently there are no available clinical trials demonstrating the influence of exercise on the gut microbiota of women with breast cancer. Nevertheless, given the benefits of exercise on the gut microbiota in healthy individuals, further research is needed to address how exercise may affect the gut microbiome of breast cancer patients. 

### 2.14. Senescent Cells 

Senescent cells do not divide but remain viable and metabolically active. Cellular senescence can be classified as (i) non-DNA damage-related (embryonic developmentally programmed senescence and physiological senescence) and (ii) DNA damage-related. Senescence signaling pathways are activated by the phosphorylation of selected proteins, such as p53/p21, p16, and Rb [162]. Indeed, p16 and p21 proteins are cell cycle inhibitors that are upregulated in senescent cells [163]. The senescence-associated secretory phenotype (SASP) contributes to immune cell recruitment and senescent cancer cell procurement [162]. Although most research highlights the role of cellular senescence as a protective mechanism against cancer, recent research shows that, in certain contexts, senescent cells can stimulate tumor development through SASP [4] by transmitting, in a paracrine way, signaling molecules to cancer cells that could activate other cancer hallmarks [4].

One study, encompassing two different experiments, did not find an increase in p53 protein in response to aerobic exercise in a murine breast cancer model [164]. Another study shows that exercise (HIIT protocol) can increase the levels of p53 protein after four weeks of training in female BALB/c mice receiving 4T1 breast cancer cell transplants [165] and reduce tumor burden through the upregulation of p53.

Cells entering senescence due to solely the exogenous overexpression of p21 and p16 proteins do not express SASP, despite requiring p53 to stop proliferation, and maintain senescent cell characteristics. On the other hand, cells lacking functional p53 protein are able to secrete elevated concentrations of many SASP components. A large number of SASP components could promote chronic inflammation, which in turn could stimulate proliferation, differentiation, angiogenesis, and metastasis [166].

The results of the aforementioned studies [164,165] are contradictory and make it difficult to understand whether exercise increases p53 protein in animal cancer models, and how it affects p53’s relationship with SASP or cellular senescence process.

Notably, it is not yet known whether exercise in breast cancer patients could act on cellular senescence mechanisms in an anti- or pro-proliferative manner (Figure 1). The recent belief that by activating SASP it is possible to promote tumor development paves the way for future research linking senescence to exercise in women with breast cancer. 

## 3. Conclusions

To the best of our knowledge, this is the first review addressing the effects of physical exercise on breast cancer hallmarks. According to the literature, the most commonly examined variables are cell proliferation, evading growth suppressors, apoptosis, immune response, and persistent inflammation. In all of these biological hallmarks, it is evident that an active and healthy lifestyle—notably including physical exercise—following diagnosis and during therapy contributes to decreased cancer severity and reduced expression of tumor-sustaining markers. On the other hand, some hallmarks (i.e., reprogramming energy metabolism, unlocking phenotypic plasticity, and senescent cells) are difficult to be studied in clinical studies, which hinders drawing consistent conclusions. There is also a lack of evidence from both preclinical and clinical studies on the effects of exercise on genomic destruction and mutation.

Concerning the promotion of angiogenesis, aerobic exercise can decrease the expression of pro-angiogenic markers, but mixed results are found for the effects of physical exercise on metastasis. Regarding replicative immortality, exercise-induced increases in telomere length correlate with a decreased tumor danger. Similarly, exercise acts on the methylation of certain genes and affects the composition of the gut microbiota [157,158], although the involved physiological mechanisms remain unknown (Figure 5).

Aerobic exercise is currently the most widely studied training approach, particularly among preclinical studies, but there are some preliminary analyses assessing the role of resistance and combined exercise. Therefore, future research—particularly clinical studies—is warranted to explore the benefits of other types of exercise.

In recent decades, a rapid progression of breast cancer exercise oncology (i.e., the study of physical exercise from the perspective of cancer prevention and management) has been observed paralleling the evolution of precision medicine. Exercise oncology research recognizes a considerable inter-individual variability in patients’ response to physical activity [167]. This viewpoint underlines the importance of precision or personalized physical activity, which can help to tailor the exercise program to the individual phenotype of the patient. Such individualized prescription is based on the individual’s functional ability, with specific recommendations regarding the dose (intensity, volume, and frequency), similarly to drug prescription [168]. Unfortunately, exercise programs are usually studied and applied with the “average person” in mind, with minor consideration for the differences between patients. Moreover, important differences between exercise and medical interventions may hamper the application of personalized medicine to exercise oncology. Much has been discovered in exercise oncology, but the application of precision physical exercise in cancer treatment is still in an early stage.

A more specific area of research is focusing on the relationships between exercise and epigenetic modifications, with the aim to enrich exercise prescription guidelines for both healthy persons and patients with different diseases. In the future, epigenetic data could increase the understanding of individual responses to specific exercise types [169]. Accordingly, physical therapists must consider that the accumulation of new epigenetic knowledge can improve the efficacy of exercise-based treatments. Continuously, examples emerge of the use of the patient’s genotype to optimize pharmacological treatments or to prevent severe drug adverse effects, which are major aims of pharmacogenomics. Tailoring exercise recommendations to the patient’s genotype will represent the next application of precision medicine. Characterization of the epigenome (i.e., the epigenetic modifications of key genes) will allow the identification of and proper treatment for patients at a high risk of cancer [169].

## Figures and Tables

**Figure 1 cancers-15-00324-f001:**
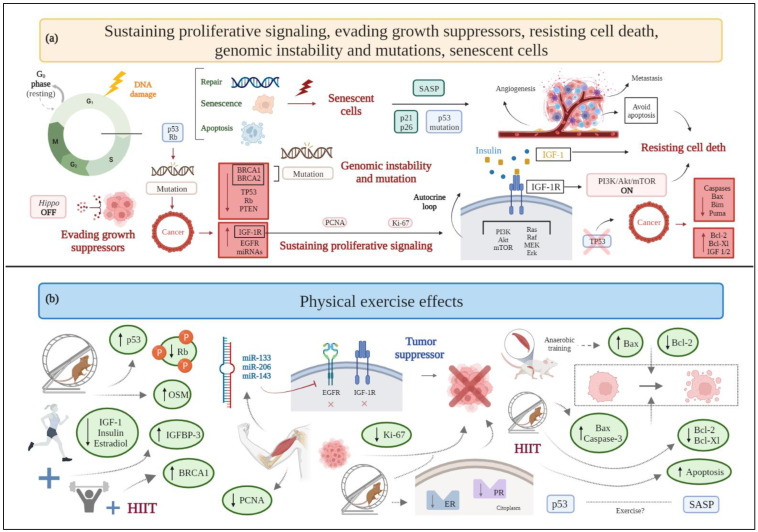
(**a**) Schematic drawing of cancer hallmarks, highlighting markers related to sustaining proliferative signaling, evading growth suppressors, resisting cell death, genomic instability, and senescent cells. (**b**) The effects of physical exercise on sustaining proliferative signaling, evading growth suppressors, resisting cell death, genomic instability, and senescent cells. Abbreviations: Akt: protein kinase b; BRCA1: breast cancer type 1; BRCA2: breast cancer type 2; EGFR: epidermal growth factor receptor; ER: estrogen receptor; Erk: extracellular signal regulated kinase; HIIT, high-intensity interval training; IGF-1: insulin-like growth factor 1; IGF-1R: insulin-like growth factor receptor type 1; IGFBP-3: insulin-like growth factor binding protein-3; miRNA: microRNA; MEK: mitogen activated protein kinase; mTOR: mammalian target of rapamycin; OSM: oncostatin M; PCNA: proliferating cell nuclear antigen; PI3K: phosphatidylinositol-3-kinase; PR: progesterone receptor; PTEN: phosphatase and tensin homolog; Rb: retinoblastoma; SASP: senescence-associated secretory phenotype; TP53: tumor protein 53.

**Figure 2 cancers-15-00324-f002:**
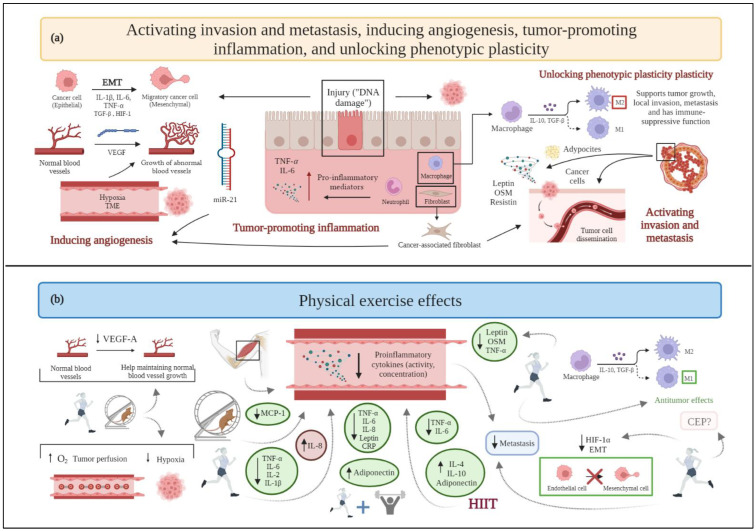
(**a**) Schematic drawing of cancer hallmarks, highlighting markers related to activating invasion and metastasis, inducing angiogenesis, tumor-promoting inflammation, and unlocking phenotypic plasticity. (**b**) The effects of physical exercise on activating invasion and metastasis, inducing angiogenesis, tumor-promoting inflammation, and unlocking phenotypic plasticity. Abbreviations: CEP: endothelial progenitor cells; EMT: epithelial to mesenchymal transition; HIF-1: hypoxia-inducible factor-1; HIF-1α: hypoxia-inducible factor-1α; HIIT: high-intensity interval training; IL-1β: interleukin-1β; IL-2: interleukin-2; Il-4: interleukin-4; IL-6: interleukin-6; IL-8: interleukin-8; IL-10: interleukin-10; MCP-1: monocyte chemoattractant protein-1; miR21: microRNA 21; OSM: oncostatin M; TGF-β: transforming growth factor-β; TNF-α: tumor necrosis factor-α; VEGF: vascular endothelial growth factor; VEGF-A: vascular endothelial growth factor-A.

**Figure 3 cancers-15-00324-f003:**
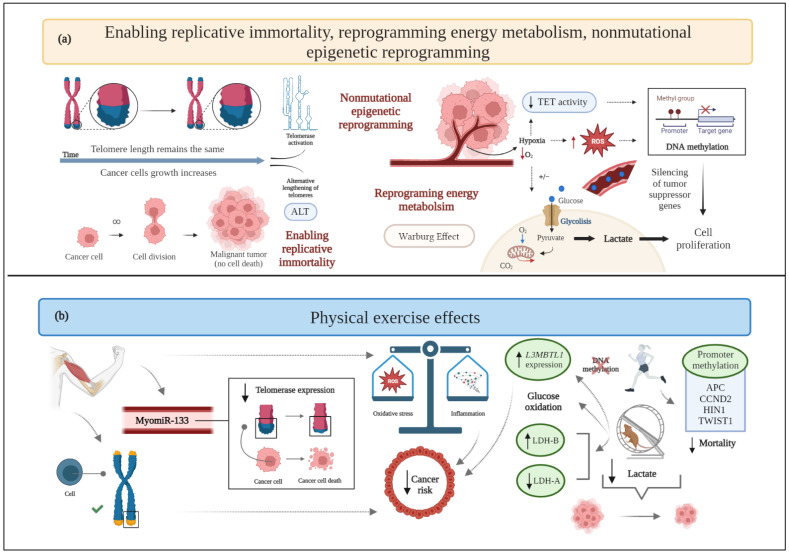
(**a**) Schematic drawing of cancer hallmarks, highlighting markers related to enabling replicative immortality, reprogramming energy metabolism, and nonmutational epigenetic reprogramming. (**b**) The effects of physical exercise on enabling replicative immortality, reprogramming energy metabolism, and nonmutational epigenetic reprogramming. ALT: alternative lengthening of telomeres; APC: adenomatous polyposis coli; CO_2_: carbon dioxide; CCND2: cyclin D2; HIN1: high in normal-1; L3MBTL1: lethal(3) malignant brain tumor-like 1; LDH-A: lactate dehydrogenase-A; LDH-B: lactate dehydrogenase-B; O_2:_ oxygen_;_ ROS: reactive oxygen species; TET: ten eleven translocation proteins; TWIST1: twist family bHLH transcription factor 1.

**Figure 4 cancers-15-00324-f004:**
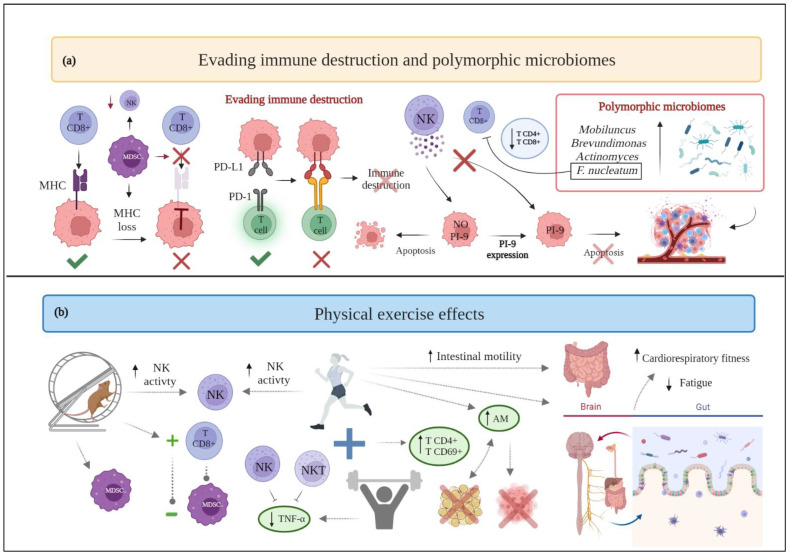
(**a**) Schematic drawing of cancer hallmarks, highlighting markers related to evading immune destruction and polymorphic microbiomes. (**b**) The effects of physical exercise on evading immune destruction and polymorphic microbiomes. Abbreviations: AM: *akkermansia muciniphilia*; F.nucleatum: Fusobacterium nucleatum; MDSC: myeloid-derived suppressor cells; MHC: major histocompatibility complex; NK: natural killer; NKT: natural killer T; PD-1: programmed death-1; PDL-1: programmed death-ligand 1; PI-9: proteinase inhibitor-9; TNF-α: Tumor Necrosis Factor-α.

**Figure 5 cancers-15-00324-f005:**
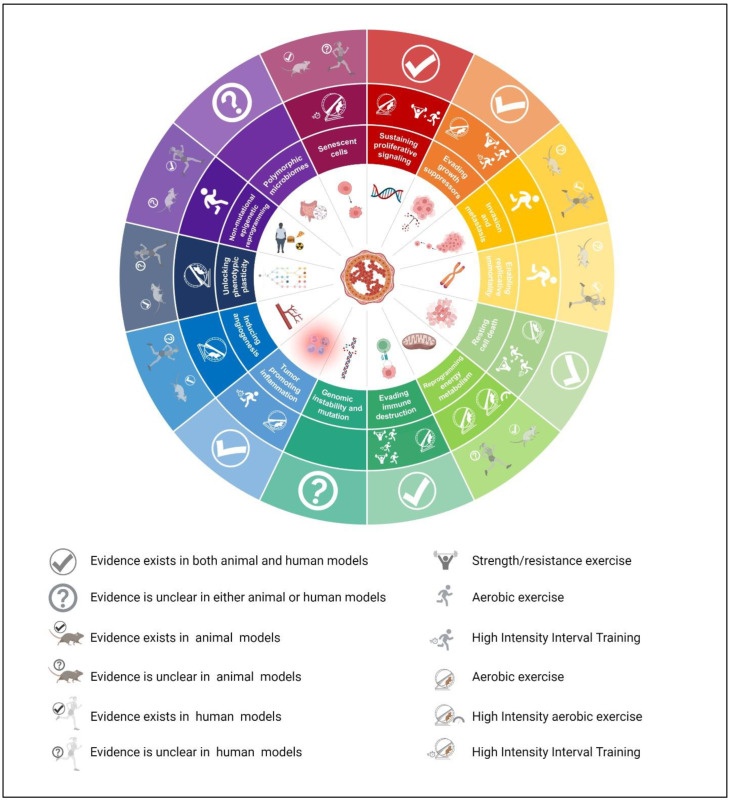
Summary of the evidence of the effects of physical exercise on the hallmarks of breast cancer.

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
