# Peer review of "Physical Exercise and the Hallmarks of Breast Cancer: A Narrative Review"

_cancers, 2023, doi:10.3390/cancers15010324_

Round 1
Reviewer 1 Report
I wish to thank the Authors for this valuable piece of research. The work is well-written, interesting, accurate and especially useful for clinicians looking for a comprehensive summary of physical activity effects on breast cancer. Paragraphs are well-organised: in each section readers are led through a brief summary and recap of tumor biology in order to better understand the subsequent information which delves deeper into biochemical mechanisms. They provide a readable but comprehensive guide for breast cancer clinicians wishing to grasp a lot of information in one go. References are punctually pointed out, in an astounding number. Figures are informative and well-drawn.
A few minor observations:
Lines 219-224: the paragraph is a little baffling as it seems to suggest that while on one hand wheel running correlates with a higher (?) number of pulmonary metastases, on the other hand (“accordingly”?) aerobic training promotes lower tumor invasiveness. Please check whether there are some typos or mistakes in the paragraph; if not, maybe the sentence could be rephrased more clearly.
Lines 245-246: in this paragraph some molecules appear in italics, in other sections molecules are introduced between quotes, while in most of the paper they are simply introduced with their abbreviation following. This can be a little confusing and I suggest choosing one method to apply throughout the paper.
Line 404: there seems to have been a problem with rogue reference 111
Line 659-661: the sentence is unclear in line 661, maybe resulting from several edits.
Figure 5: great figure! It provides an excellent recap of a complex paper (and great potential slide material!). However, the question mark is a little unclear: does it indicate unclear evidence in “either” animal or human models or in both animal “and” human models?
Author Response
I wish to thank the Authors for this valuable piece of research. The work is well-written, interesting, accurate and especially useful for clinicians looking for a comprehensive summary of physical activity effects on breast cancer. Paragraphs are well-organised: in each section readers are led through a brief summary and recap of tumor biology in order to better understand the subsequent information which delves deeper into biochemical mechanisms. They provide a readable but comprehensive guide for breast cancer clinicians wishing to grasp a lot of information in one go. References are punctually pointed out, in an astounding number. Figures are informative and well-drawn.
Comments much appreciated. Additions, as well as corrections to the original manuscript are marked (red font) in the revised manuscript.
- Lines 219-224: the paragraph is a little baffling as it seems to suggest that while on one hand wheel running correlates with a higher (?) number of pulmonary metastases, on the other hand (“accordingly”?) aerobic training promotes lower tumor invasiveness. Please check whether there are some typos or mistakes in the paragraph; if not, maybe the sentence could be rephrased more clearly.
- Thanks for the comment. The indicated sentence has been corrected.
- Lines 245-246: in this paragraph some molecules appear in italics, in other sections molecules are introduced between quotes, while in most of the paper they are simply introduced with their abbreviation following. This can be a little confusing and I suggest choosing one method to apply throughout the paper.
- In the revised manuscript, italics was used only for abbreviations referring to genes. Moreover, molecules named between quotes have been corrected.
- Line 404: there seems to have been a problem with rogue reference 111
- Thanks for noticing. There was an error with the reference. Our apologies.
- Line 659-661: the sentence is unclear in line 661, maybe resulting from several edits.
- Thanks for the comment. We have rephrased the sentence to make it more understandable.
- Figure 5: great figure! It provides an excellent recap of a complex paper (and great potential slide material!). However, the question mark is a little unclear: does it indicate unclear evidence in “either” animal or human models or in both animal “and” human models?
- Thanks for the comment. We have corrected the sentence ("in either animal or human models")
Reviewer 2 Report
Dear Authors,
As epidemiological data confirm, breast cancer is the most frequent neoplasm. However, the most recent advancements in available therapeutical options have achieved to dramatically increase survival rates, thus highlighting the need of tailored chronic management and prevention strategies.
As a result, current literature is now addressing the role of physical exercise both for prevention and slowing cancer’s progression.
In this scenario, I think that your manuscript outstandingly depicts the state of art on this field of research, brilliantly addressing the role of different physical exercise protocols and showing how even exercise intensity could potentially impact on cancer hallmarks.
Indeed, as genetical and molecular pathways involved are so well-characterized, I think that your manuscript would gain even more quality by addressing some minor reviews provided below.
Minor reviews
SIMPLE SUMMARY: Page 1, line 18, please, correct “increase” with “increases”. Page 1, line 21, please, correct “generate” with “generates”.
INTRODUCTION: The whole section is focused and well written. However, in the complexity of breast cancer treatments, potential cardio-toxic effects of chemotherapy should be addressed, because cardio-respiratory fitness might be impaired. Indeed, there is shortage of literature regarding this topic; however, you might consider some references here enclosed:
- Invernizzi M, Lippi L, Folli A, Turco A, Zattoni L, Maconi A, de Sire A, Fusco N. Integrating molecular biomarkers in breast cancer rehabilitation. What is the current evidence? A systematic review of randomized controlled trials. Front Mol Biosci. 2022 Sep 8;9:930361. doi: 10.3389/fmolb.2022.930361.
- Maginador G, Lixandrão ME, Bortolozo HI, Vechin FC, Sarian LO, Derchain S, Telles GD, Zopf E, Ugrinowitsch C, Conceição MS. Aerobic Exercise-Induced Changes in Cardiorespiratory Fitness in Breast Cancer Patients Receiving Chemotherapy: A Systematic Review and Meta-Analysis. Cancers (Basel). 2020 Aug 11;12(8):2240. doi: 10.3390/cancers12082240.
- Ma ZY, Yao SS, Shi YY, Lu NN, Cheng F. Effect of aerobic exercise on cardiotoxic outcomes in women with breast cancer undergoing anthracycline or trastuzumab treatment: a systematic review and meta-analysis. Support Care Cancer. 2022 Nov 2. doi: 10.1007/s00520-022-07368-w.
EVADING GROWTH SUPPRESSORS: Page 4, line 139. Please, put reference number in square brackets (i.e. (33)).
EVADING IMMUNE DISRUPTION: Page 10, line 404. It seems there is a typo (i.e. [111][111]). Please, correct it.
FIGURES: Please, note that figures should be structured to stand independently from the manuscript; hence, captions should be implemented by providing the list of the abbreviations used.
FIGURE 4: The figure is well structured and focused. However, I think that it might be improved as here suggested:
- As addressing the main focus of the manuscript, the external ring, together with its content, should be enlarged.
- It should be added a second ring reporting whether physical activity protocols assessed in the studies included in your research are classified as aerobic, resistance, combined, or high interval training.
- The text within the actual second ring, which would become the third ring, should be centered in relation to the height of the ring.
Author Response
Dear Authors,
As epidemiological data confirm, breast cancer is the most frequent neoplasm. However, the most recent advancements in available therapeutical options have achieved to dramatically increase survival rates, thus highlighting the need of tailored chronic management and prevention strategies.
As a result, current literature is now addressing the role of physical exercise both for prevention and slowing cancer’s progression.
In this scenario, I think that your manuscript outstandingly depicts the state of art on this field of research, brilliantly addressing the role of different physical exercise protocols and showing how even exercise intensity could potentially impact on cancer hallmarks.
Indeed, as genetical and molecular pathways involved are so well-characterized, I think that your manuscript would gain even more quality by addressing some minor reviews provided below.
Comments much appreciated. Additions, as well as corrections to the original manuscript are marked (red font) in the revised manuscript.
Minor reviews
- SIMPLE SUMMARY: Page 1, line 18, please, correct “increase” with “increases”. Page 1, line 21, please, correct “generate” with “generates”.
- Thanks for noticing. Corrections have been done as recommended.
- INTRODUCTION: The whole section is focused and well written. However, in the complexity of breast cancer treatments, potential cardio-toxic effects of chemotherapy should be addressed, because cardio-respiratory fitness might be impaired. Indeed, there is shortage of literature regarding this topic; however, you might consider some references here enclosed:
- Invernizzi M, Lippi L, Folli A, Turco A, Zattoni L, Maconi A, de Sire A, Fusco N. Integrating molecular biomarkers in breast cancer rehabilitation. What is the current evidence? A systematic review of randomized controlled trials. Front Mol Biosci. 2022 Sep 8;9:930361. doi: 10.3389/fmolb.2022.930361.
- Maginador G, Lixandrão ME, Bortolozo HI, Vechin FC, Sarian LO, Derchain S, Telles GD, Zopf E, Ugrinowitsch C, Conceição MS. Aerobic Exercise-Induced Changes in Cardiorespiratory Fitness in Breast Cancer Patients Receiving Chemotherapy: A Systematic Review and Meta-Analysis. Cancers (Basel). 2020 Aug 11;12(8):2240. doi: 10.3390/cancers12082240.
- Ma ZY, Yao SS, Shi YY, Lu NN, Cheng F. Effect of aerobic exercise on cardiotoxic outcomes in women with breast cancer undergoing anthracycline or trastuzumab treatment: a systematic review and meta-analysis. Support Care Cancer. 2022 Nov 2. doi: 10.1007/s00520-022-07368-w.
- We thank the reviewer for the valuable and stimulating comment. In the revised manuscript, the suggested references have been cited and a brief discussion has been added concerning physical exercise effects on cardiorespiratory fitness and on chemotherapy-related cardiotoxicity.
- EVADING GROWTH SUPPRESSORS: Page 4, line 139. Please, put reference number in square brackets (i.e. (33)).
- Thanks for noticing. There was an error with the reference. Our apologies.
- EVADING IMMUNE DISRUPTION: Page 10, line 404. It seems there is a typo (i.e. [111][111]). Please, correct it.
- Thanks for noticing. There was an error with the reference.
- FIGURES: Please, note that figures should be structured to stand independently from the manuscript; hence, captions should be implemented by providing the list of the abbreviations used.
- Thanks for the comment. We have included a list of abbreviations below each figure.
- FIGURE 4: The figure is well structured and focused. However, I think that it might be improved as here suggested:
- As addressing the main focus of the manuscript, the external ring, together with its content, should be enlarged.
- It should be added a second ring reporting whether physical activity protocols assessed in the studies included in your research are classified as aerobic, resistance, combined, or high interval training.
- The text within the actual second ring, which would become the third ring, should be centered in relation to the height of the ring.
- Thanks for the comment. We have made changes to the figure to improve its understanding.